# Estimating the Associated Burden of Illness and Healthcare Utilization of Newly Diagnosed Patients Aged ≥65 with Mantle Cell Lymphoma (MCL) in Ontario, Canada

Peter Anglin [1,2], Julia Elia-Pacitti [3], Maria Eberg [4], Sergey Muratov [5], Atif Kukaswadia [5], Arushi Sharma [6] and Emmanuel M. Ewara [3,*]

1  Stronach Regional Cancer Centre, Southlake Regional Health Centre, Newmarket, ON L3Y 2P9, Canada; panglin@southlakeregional.org
2  Bayshore HealthCare, 2101 Hadwen Rd., Mississauga, ON L5K 2L3, Canada
3  Janssen Canada Inc., 19 Green Belt Drive, North York, ON M3C 1L9, Canada; jpacitti@its.jnj.com
4  Real World Solutions, IQVIA, 16720 Rte Transcanadienne, Kirkland, QC H9H 5M3, Canada; masha.eberg@iqvia.com
5  Real World Solutions, IQVIA, 402-1875 Buckhorn Gate, Mississauga, ON L4W 5N9, Canada; sergey.muratov@iqvia.com (S.M.); atif.kukaswadia@iqvia.com (A.K.)
6  Real World Solutions, IQVIA, 301-300 March Rd., Kanata, ON K2K 2E2, Canada; arushi.fraelic@iqvia.com
*  Correspondence: eewara@its.jnj.com

**Abstract:** Background: With the emergence of therapies for mantle cell lymphoma (MCL), understanding the treatment patterns and burden of illness among older patients with MCL in Canada is essential to inform decision making. Methods: A retrospective study using administrative data matched individuals aged ≥65 who were newly diagnosed with MCL between 1 January 2013 and 31 December 2016 with general population controls. Cases were followed for up to 3 years in order to assess healthcare resource utilization (HCRU), healthcare costs, time to next treatment or death (TTNTD), and overall survival (OS); all were stratified according to first-line treatment. Results: This study matched 159 MCL patients to 636 controls. Direct healthcare costs were highest among MCL patients in the first year following diagnosis (Y1: CAD 77,555 ± 40,789), decreased subsequently (Y2: CAD 40,093 ± 28,720; Y3: CAD 36,059 ± 36,303), and were consistently higher than the costs for controls. The 3-year OS after MCL diagnosis was 68.6%, with patients receiving bendamustine + rituximab (BR) experiencing a significantly higher OS compared to patients treated with other regimens (72.4% vs. 55.6%, $p = 0.041$). Approximately 40.9% of MCL patients initiated a second-line therapy or died within 3 years. Conclusion: Newly diagnosed MCL presents a substantial burden to the healthcare system, with almost half of all patients progressing to a second-line therapy or death within 3 years.

**Keywords:** burden of illness; mantle cell lymphoma; costs; epidemiology; healthcare utilization

## 1. Introduction

Non-Hodgkin's lymphomas (NHL) are a heterogenous group of neoplasms originating from the lymphoid tissues [1], of which mantle cell lymphoma (MCL) constitutes 3–10% of all cases. There is an estimated 500–600 newly diagnosed MCL cases per year in Canada [2,3], mainly arising in older adults (median age 60–65 years) [4]. While approximately 30% of MCL patients demonstrate a more indolent course [5], the majority endure advanced-stage disease that follows an aggressive course [4,6]. Accordingly, MCL is associated with poor long-term survival, with a median overall survival (OS) of approximately 3–5 years [4,7]; in addition, advanced age is associated with decreased survival. In line with this, the 5-year survival for patients >75 years of age has been reported as 17%, compared with 78% for patients <40 years of age [8].

The recommended first-line treatment strategies for MCL differ according to the age and fitness of the patients [7]. Currently, newly diagnosed MCL patients <65–70 years of age who do not present with significant comorbidities receive chemoimmunotherapy (induction) and, if responsive, a subsequent high-dose therapy followed by an autologous stem cell transplantation (ASCT; consolidation), with rituximab maintenance thereafter [6]. Commonly used chemoimmunotherapy regimens include rituximab-based regimens such as R-CHOP (rituximab, cyclophosphamide, doxorubicin, vincristine, and prednisone) or alternating R-CHOP/R-DHAP (rituximab, dexamethasone, high-dose cytarabine, and cisplatin). Occasionally, hyper-CVAD (hyperfractionated cyclophosphamide, vincristine, doxorubicin, dexamethasone, alternating with methotrexate and cytarabine) is adopted [3,9–11]. As patients aged >65–70 years are typically ineligible for transplant, management focuses on less toxic treatment strategies such as bendamustine and rituximab (BR) and R-CHOP [6,9]. However, as bendamustine was not available in Canada until 2012 [12], its availability as a treatment for MCL patients may not have been observed until some years later, and access may differ depending on jurisdictional funding status.

Cancer presents a substantial burden on the Canadian population and the healthcare system [13]. Currently, evidence regarding the economic burden of MCL remains limited, but a considerable variation in costs according to the treatment regimen and care setting has been demonstrated [8,14]. Adverse events (AE) were found to be the key drivers of increased costs and resource use in the USA, largely through inpatient admissions and outpatient visits [14]. However, these studies included more economic evaluations rather than reports on real-world resource use and humanistic burden [8], indicating the need for further research in order to assess the true economic burden of this disease, especially in a Canadian setting where evidence remains limited.

The primary objective of this study was to investigate the real-world burden of illness (BOI) of newly diagnosed Canadian patients with MCL aged ≥65, with a focus on those in the first line of treatment (LoT) with systemic therapies. The secondary objectives were to describe the treatment patterns, healthcare resource utilization (HCRU), direct healthcare costs, overall survival (OS), and time to next treatment or death (TTNTD) of patients aged ≥65 with newly diagnosed MCL.

## 2. Methods

### 2.1. Study Design

A retrospective, longitudinal, population-based study was conducted using administrative health data from Ontario, Canada. A patient's diagnosis was determined using the International Classification of Disease for Oncology (ICD-O-3) morphology code for MCL (96733). Individuals aged ≥65 and newly diagnosed with MCL between 1 January 2013 and 31 December 2016 were included. Patients aged ≥65 qualify for the provincial publicly funded drug program (Ontario Drug Benefit; ODB). Focusing our analysis on this age group allowed a more fulsome perspective of healthcare costs and utilization. Patients were followed for up to three years post-diagnosis (index event). Demographic and clinical characteristics for each patient were ascertained from administrative records prior to MCL diagnosis.

Patients had to have received systemic therapy within three years following MCL diagnosis to be included in the study cohort. MCL patients were excluded if they met any of the following criteria: an invalid public health coverage (Ontario Health Insurance Plan; OHIP), invalid or incomplete records (i.e., missing age, sex, other demographic information, age ≥105 at MCL diagnosis, death on date of diagnosis), primary residency outside of Ontario, or received multiple incident cancer diagnoses on the same date. Patients were also excluded if they received ASCT at any time before the end of the study follow-up period, initiated a regimen outside of the recommended first-line treatments for MCL (Supplementary Table S1), had participated in a clinical trial, or had refractory disease or treatment intolerance.

### 2.2. Data Sources

In Ontario, all residents receive medically necessary care under a single payer insurance system (OHIP). Healthcare utilization information is available in the form of deidentified administrative records that are linked using uniquely encoded identifiers housed at the Institute for Clinical and Evaluative Sciences (ICES), a not-for-profit research institute whose legal status under Ontario's health information privacy law allows it to collect and analyze healthcare data for health system evaluation and improvement. The Ontario Cancer Registry (OCR) was used to identify patients diagnosed with MCL, their cancer diagnosis date, and prior cancer history.

Demographic data, including sex, age, and residential postal codes, were extracted from the Registered Persons Database (RPDB). The neighborhood-level income quintile was derived using census data, based on the median income in each dissemination area [15], and linked back to patients using their residential postal codes [16]. Comorbidities were defined using diagnosis records from the Discharge Abstract Database (DAD), Same Day Surgery (SDS), National Ambulatory Care Reporting System (NACRS), and OHIP databases, with two records required to flag a comorbidity using the OHIP database. Where feasible, ICES-derived cohorts [17] were used to identify comorbid conditions such as diabetes mellitus, congestive heart failure, and chronic obstructive pulmonary disease (COPD). The Charlson–Deyo comorbidity score was assigned based on DAD records in the two years prior to the index date. Treatment information was obtained from the Cancer Activity Level Reporting (ALR) datasets. ODB data captured all prescription claims dispensed under Ontario's provincial public drug program. The DAD, NACRS, and OHIP data were used to estimate HCRU. All costs were estimated using a person-centered costing methodology developed at ICES [18] using data sources that included the following: DAD, SDS, NACRS, OHIP, and New Drug Funding Program (NDFP).

### 2.3. Selection of General Population Controls

To estimate the utilization and costs attributable to MCL, each MCL patient who initiated a first LoT during the study follow-up was matched to up to four general population controls. An individual was eligible as a general population control if they had never been diagnosed with MCL as per the OCR, were aged ≥65 on the index date, had primary residency within Ontario, and had valid healthcare records. Matching was performed on the following characteristics: index date (±90 days), age ≥65 at index (±5 years), sex (exact match), history of any cancer within 5 years before index (exact match), and Local Health Integration Network (LHIN; exact match). LHIN is the geographic partition of Ontario into healthcare regions, where each region has its own set of healthcare providers and its own administration to coordinate care and distribute funds; exact matching on LHIN ensures that cases and controls have similar access to healthcare facilities and services.

### 2.4. Study Outcomes

HCRU was assessed at various healthcare touchpoints, including general practitioner/family physician (GP/FP) visits, oncologist and hematologist visits, all other specialist visits, hospitalizations, and emergency department (ED) visits. Total direct healthcare costs were estimated as the weighted sum of GP/FP costs, oncologist and hematologist costs, other specialist costs, hospitalization costs, same-day surgeries, ED costs, cancer clinic costs, ODB costs, NDFP chemotherapy drug costs, and other healthcare costs for each year over a patient's follow-up period. Other healthcare costs included the direct costs of dialysis clinics, rehabilitation services, complex and continuing care, home care services, laboratory test billings, non-physician billings, shadow billings, primary care physician capitation, long-term care, mental health care admissions, assisted devices, and outpatient hospital clinic visits. Estimated amounts were standardized to 2020 Canadian dollars.

OS was measured over time by examining the date of death, as reported in the RPDB, and analyzed using a cumulative incidence function (CIF). TTNTD was measured as the time from the first administration date of the first LoT until the first administration date of

the second LoT or death date, as registered in RPDB. Censoring was defined as the end of the analysis period, end of OHIP coverage, or 31 December 2019, whichever occurred first. The incidence of TTNTD was also analyzed using a CIF.

### 2.5. Stratification

The overall pool of newly diagnosed MCL patients was stratified into subgroups based on the systemic therapies received as the first LoT. Subgroups were defined as patients receiving BR vs. other regimens, which included R-CHOP, R-CVP, CHLO, and FCR. HCRU, costs, OS, and TTNTD analyses were performed separately for the first LoT subgroups.

### 2.6. Statistical Analysis

Descriptive analyses (frequency, mean, median) were calculated to summarize the demographic characteristics for both patient cohorts. Absolute standardized differences were used to assess the balance in the distribution of the baseline characteristics between MCL patients and their matched controls. Annual estimates of HCRU counts and total costs were estimated as means by healthcare touchpoint per patient. To account for the matched nature of the study, a generalized estimating equations (GEE) analysis was employed. Data were pulled in September 2021 and analyzed using SAS Enterprise Guide v7.15 (SAS Institute Inc., Cary, NC, USA). All analyses were conducted by ICES staff. Small cell values (strata < 6) were reported as a range (1–5) in accordance with ICES reporting standards.

### 2.7. Ethics

This study was designed and implemented with ethics approval from the Institutional Review Board Services (Advarra IRB# 00000971, Approval #IBR-C-21-CAN-002-V01/2722439), and was approved by the ICES Privacy and Compliance Office.

## 3. Results

### 3.1. Study Population

A total of 313 newly diagnosed cases of MCL in patients aged ≥65 were initially identified between 1 January 2013 and 31 December 2016. Twelve patients were excluded due to the receipt of ASCT within three years of MCL diagnosis. In total, 154 (49.2%) patients with MCL were excluded because they had no record of systemic therapy within three years following MCL diagnosis, had received a first LoT outside of the confirmed treatments for MCL (Supplemental Table S1), had refractory disease, treatment intolerance, or could not be matched to four controls. After all exclusion criteria were applied, our final cohort consisted of 159 patients (Supplemental Table S2).

### 3.2. Patient Characteristics

The demographic and clinical characteristics of the MCL cohort and their matched controls are presented in Table 1. The MCL cohort was 67.9% (N = 108) male, with a median age of 75 (IQR: 70–80) at the time of diagnosis, and 9.4% (N = 15) had a history of cancer within five years prior to MCL diagnosis. The majority of the cohort (84.9%) lived in large urban centres, while there was an even distribution of patients across income quintiles. The most common therapy received as a first LoT was BR (78%), followed by R-CHOP (13%) (Supplemental Table S3). Patients on BR comprised a higher proportion of males (71.5%) compared with those on other regimens (55.6%) (Supplemental Table S4).

### 3.3. Healthcare Resource Utilization

In the three years after diagnosis, >95% of patients had at least one attending physician, defined as the specialty of a physician with whom the patient had the highest number of visits within a 12-month period. The most common attending physician for the MCL cohort was a hematologist in years one (45.9%) and two (41.4%), and a GP/FP in year three (44.8%). On the other hand, most patients in the general control population had a GP/FP as their attending physician throughout all three years of follow-up (63.5–65.2%).

**Table 1.** Demographic and clinical characteristics of MCL patients and their matched controls.

| Demographic and Clinical Characteristics | | MCL Patients N = 159 | Controls N = 636 | Absolute Standardized Difference |
|---|---|---|---|---|
| Sex | Female | 51 (32.1%) | 204 (32.1%) | 0.00 |
| | Male | 108 (67.9%) | 432 (67.9%) | 0.00 |
| Age | Mean, SD | 75.3 ± 6.2 | 74.7 ± 6.5 | 0.09 |
| | Median (IQR) | 75 (70–80) | 74 (69–80) | n.a. |
| | Min–Max | 65–92 | 65–96 | n.a. |
| | 65–74 years | 77 (48.4%) | 331 (52%) | 0.07 |
| | 75+ years | 82 (51.6%) | 305 (48%) | 0.07 |
| Rural residence | Urban | 135 (84.9%) | 547 (86%) | 0.03 |
| | Rural | 24 (15.1%) | 89 (14%) | 0.03 |
| Income quintile | Q1, lowest | 24 (15.1%) | 121 (19%) | 0.10 |
| | Q2 | 33 (20.8%) | 131 (20.6%) | 0.00 |
| | Q3 | 29 (18.2%) | 117 (18.4%) | 0.00 |
| | Q4 | 33 (20.8%) | 131 (20.6%) | 0.00 |
| | Q5, highest | 40 (25.2%) | 136 (21.4%) | 0.09 |
| New Ontario resident at diagnosis | | * 1–5 | * 10–14 | 0.00 |
| Local Health Integration Network | 1. Erie St. Clair | 9 (5.7%) | 36 (5.7%) | 0.00 |
| | 2. South West | 9 (5.7%) | 36 (5.7%) | 0.00 |
| | 3. Waterloo Wellington | 10 (6.3%) | 40 (6.3%) | 0.00 |
| | 4. Hamilton Niagara Haldimand Brant | 15 (9.4%) | 60 (9.4%) | 0.00 |
| | 5. Central West | 7 (4.4%) | 28 (4.4%) | 0.00 |
| | 6. Mississauga Halton | 7 (4.4%) | 28 (4.4%) | 0.00 |
| | 7. Toronto Central | 17 (10.7%) | 68 (10.7%) | 0.00 |
| | 8. Central | 19 (11.9%) | 76 (11.9%) | 0.00 |
| | 9. Central East | 21 (13.2%) | 84 (13.2%) | 0.00 |
| | 10. South East | 8 (5%) | 32 (5%) | 0.00 |
| | 11. Champlain | 13 (8.2%) | 52 (8.2%) | 0.00 |
| | 12. North Simcoe Muskoka | 7 (4.4%) | 28 (4.4%) | 0.00 |
| | 13. North East | 11 (6.9%) | 44 (6.9%) | 0.00 |
| | 14. North West | 6 (3.8%) | 24 (3.8%) | 0.00 |
| Cancer history (assessed within a 5-year lookback) ^ | Any cancer | 15 (9.4%) | 60 (9.4%) | 0.00 |

**Table 1.** *Cont.*

| Demographic and Clinical Characteristics | | MCL Patients N = 159 | Controls N = 636 | Absolute Standardized Difference |
|---|---|---|---|---|
| Charlson co-morbidity index (assessed within a 2-year lookback) | 0 | 23 (14.5%) | 65 (10.2%) | 0.13 |
| | 1 | 7 (4.4%) | 18 (2.8%) | 0.08 |
| | 2 | * 2–6 | * 16–20 | 0.03 |
| | 3+ | * 1–5 | * 16–20 | 0.04 |
| | Missing | 119 (74.8%) | 520 (81.8%) | 0.17 |
| Comorbidities, any time before index date (ICES-derived cohorts) | Diabetes mellitus | 38 (23.9%) | 197 (31%) | 0.16 |
| | Congestive heart failure (CHF) | 9 (5.7%) | 56 (8.8%) | 0.12 |
| | Chronic obstructive pulmonary disease (COPD) | 35 (22%) | 130 (20.4%) | 0.04 |
| | Rheumatoid arthritis (RA) | * 1–5 | * 20–24 | 0.15 |
| | Crohn's/Colitis | * 1–5 | * 1–5 | 0.08 |
| Time from index date to end of follow-up, days | Mean, SD | 915.3 ± 321.5 | 1,029.7 ± 207.9 | 0.42 |
| | Median (IQR) | 1,095 (841–1,095) | 1,095 (1,095–1,095) | n.a. |
| | Min–Max | 28–1,095 | 66–1,095 | n.a. |

* Double suppression was conducted according to ICES reporting standards to reduce the risk of patient re-identification. ˆ Due to small cell sizes, the types of previous cancers could not be reported. n.a.—not available.

In the first year following diagnosis, MCL patients interacted with the healthcare system in a myriad of ways: >95% made at least one visit to a GP/FP, >60% visited the ED, and ~50% were hospitalized (Table 2). Throughout the follow-up period, HCRU was significantly higher ($p < 0.05$) for MCL patients compared to controls in terms of the average number of visits (Table 2).

**Table 2.** Distribution of outcomes for MCL patients and their controls.

| BOI Outcomes | Year 1 | | | Year 2 | | | Year 3 | | |
|---|---|---|---|---|---|---|---|---|---|
| Number of Patients | MCL Patients | Controls | *p*-Value | MCL Patients | Controls | *p*-Value | MCL Patients | Controls | *p*-Value |
| | *N = 159* | *N = 636* | | *N = 140* | *N = 611* | | *N = 125* | *N = 581* | |
| Person years | | | | | | | | | |
| Mean, SD | 0.94 ± 0.19 | 0.98 ± 0.11 | n.a. | 0.94 ± 0.18 | 0.98 ± 0.12 | n.a. | 0.94 ± 0.20 | 0.99 ± 0.09 | n.a. |
| Median, IQR | 1 (1–1) | 1 (1–1) | | 1 (1–1) | 1 (1–1) | | 1 (1–1) | 1 (1–1) | |
| **Healthcare resource utilization (HCRU)** | | | | | | | | | |
| Attending physician specialty | | | | | | | | | |
| Medical oncologist | 15 (9.4%) | * 1–5 | n.a. | * 10–14 | * 1–5 | n.a. | * 6–10 | * 1–5 | n.a. |
| Hematologist | 73 (45.9%) | * 1–5 | n.a. | 58 (41.4%) | * 1–5 | n.a. | 36 (28.8%) | * 1–5 | n.a. |
| Internal medicine | 24 (15.1%) | 26 (4.1%) | n.a. | 11 (7.9%) | 31 (5.1%) | n.a. | 10 (8%) | 23 (4%) | n.a. |
| Family practice/GP | 33 (20.8%) | 404 (63.5%) | n.a. | 46 (32.9%) | 396 (64.8%) | n.a. | 56 (44.8%) | 379 (65.2%) | n.a. |

**Table 2.** *Cont.*

| BOI Outcomes | Year 1 | | | Year 2 | | | Year 3 | | |
|---|---|---|---|---|---|---|---|---|---|
| Number of Patients | MCL Patients | Controls | *p*-Value | MCL Patients | Controls | *p*-Value | MCL Patients | Controls | *p*-Value |
| Other | 14 (8.8%) | 157 (24.7%) | n.a. | 13 (9.3%) | 134 (21.9%) | n.a. | 12 (9.6%) | 127 (21.9%) | n.a. |
| No attending physician | 0 (0%) | 42 (6.6%) | n.a. | * 1–5 | * 43–47 | n.a. | * 1–5 | * 45–49 | n.a. |
| | | | | | | | | | |
| **GP visits** | | | | | | | | | |
| Mean, SD | 12.4 ± 15.7 | 7.9 ± 10.0 | <0.0001 | 9.8 ± 10.4 | 7.9 ± 10.4 | 0.0467 | 10.9 ± 13.1 | 8.4 ± 13.0 | 0.0377 |
| Median, IQR | 8 (5–15) | 5 (2–10) | | 6.5 (4–12.5) | 5 (2–10) | | 7 (3–14) | 5 (2–10) | |
| **Oncologist and hematologist visits** | | | | | | | | | |
| Mean, SD | 18.0 ± 14.8 | 0.4 ± 3.5 | <0.0001 | 10.8 ± 11.1 | 0.4 ± 2.5 | <0.0001 | 8.3 ± 9.6 | 0.3 ± 2.0 | <0.0001 |
| Median, IQR | 18 (5–26) | 0 (0–0) | | 8 (5–13) | 0 (0–0) | | 5 (2–9) | 0 (0–0) | |
| **Other specialist visits** | | | | | | | | | |
| Mean, SD | 27.0 ± 24.5 | 9.7 ± 14.0 | <0.0001 | 18.0 ± 21.0 | 9.8 ± 15.0 | <0.0001 | 18.2 ± 21.0 | 9.4 ± 13.9 | <0.0001 |
| Median, IQR | 21 (11–35) | 5 (2–12) | | 11 (5–21.5) | 5 (2–12) | | 12 (5–22) | 5 (2–10) | |
| **Inpatient hospitalizations** | | | | | | | | | |
| Mean, SD | 0.8 ± 1.2 | 0.2 ± 0.7 | <0.0001 | 0.6 ± 1.2 | 0.2 ± 0.5 | <0.0001 | 0.5 ± 0.9 | 0.2 ± 0.6 | <0.0001 |
| Median, IQR | 0 (0–1) | 0 (0–0) | | 0 (0–1) | 0 (0–0) | | 0 (0–1) | 0 (0–0) | |
| **Emergency department visits** | | | | | | | | | |
| Mean, SD | 1.5 ± 1.8 | 0.6 ± 1.3 | <0.0001 | 1.3 ± 2.0 | 0.6 ± 1.3 | <0.0001 | 1.0 ± 1.5 | 0.6 ± 1.3 | 0.0090 |
| Median, IQR | 1 (0–2) | 0 (0–1) | | 0 (0–2) | 0 (0–1) | | 0 (0–2) | 0 (0–1) | |
| | | | | | | | | | |
| **Direct healthcare costs** | | | | | | | | | |
| Total direct costs | | | | | | | | | |
| Mean, SD | 77,554.7 ± 40,788.5 | 11,123.8 ± 24,515.1 | <0.0001 | 40,093.0 ± 28,719.7 | 10,891.1 ± 24,060.6 | <0.0001 | 36,059.2 ± 36,302.9 | 11,233.9 ± 26,261.2 | <0.0001 |
| Median, IQR | 83,931.0 (50,942.0–105,899.0) | 3,105.0 (1,234.5–8,567.5) | | 31,853.5 (25,054.5–54,577.5) | 2,974.0 (1,200.0–8,884.0) | | 23,500.0 (14,945.0–40,162.0) | 2,589.0 (1,159.0–6,743.0) | |
| GP costs | | | | | | | | | |
| Mean, SD | 463.1 ± 807.6 | 281.6 ± 526.7 | 0.0022 | 437.7 ± 808.2 | 280.9 ± 548.7 | 0.0135 | 475.2 ± 927.7 | 301.5 ± 751.8 | 0.0266 |
| Median, IQR | 191.0 (19.0–529.0) | 101.0 (3.5–306.5) | | 175.0 (11.0–450.0) | 88.0 (0–286.0) | | 127.0 (10.0–452.0) | 82.0 (0–273.0) | |
| Oncologist and hematologist costs | | | | | | | | | |
| Mean, SD | 907.3 ± 820.8 | 170.5 ± 132.8 | <0.0001 | 309.2 ± 392.2 | 213.7 ± 177.6 | <0.0001 | 259.5 ± 392.2 | 222.0 ± 184.4 | <0.0001 |
| Median, IQR | 860.0 (86.0–1,556.0) | 127.0 (91.0–254.0) | | 203.0 (0–505.0) | 174.0 (123.0–285.0) | | 81.0 (0–296.0) | 170.0 (103.0–264.0) | |
| Other specialist costs | | | | | | | | | |
| Mean, SD | 3,349.4 ± 2,575.7 | 1,427.3 ± 2,085.8 | <0.0001 | 2,194.6 ± 2,449.3 | 1,326.0 ± 1,806.2 | <0.0001 | 2,324.3 ± 2,651.9 | 1,244.7 ± 1,719.2 | <0.0001 |
| Median, IQR | 2,674.0 (1,687.0–4,082.0) | 715.0 (240.0–1,682.0) | | 1,212.0 (570.0–2,883.0) | 719.0 (231.0–1,672.0) | | 1,329.0 (568.0–2,628.0) | 666.0 (241.0–1,343.0) | |

**Table 2.** *Cont.*

| BOI Outcomes | Year 1 | | | Year 2 | | | Year 3 | | |
|---|---|---|---|---|---|---|---|---|---|
| Number of Patients | MCL Patients | Controls | *p*-Value | MCL Patients | Controls | *p*-Value | MCL Patients | Controls | *p*-Value |
| Inpatient hospitalization costs | | | | | | | | | |
| Mean, SD | 8,953.6 ± 19,016.9 | 2,367.3 ± 8,420.3 | <0.0001 | 6,037.6 ± 16,122.6 | 2,611.9 ± 9,147.3 | 0.0012 | 5,956.4 ± 13,823.6 | 3,022.2 ± 14,013.1 | 0.0319 |
| Median, IQR | 0 (0– 11,472.0) | 0 (0–0) | | 0 (0–5,539.0) | 0 (0–0) | | 0 (0–6,799.0) | 0 (0–0) | |
| Same-day surgery costs | | | | | | | | | |
| Mean, SD | 839.8 ± 1,155.6 | 330.2 ± 1,036.3 | <0.0001 | 384.2 ± 1,219.4 | 299.6 ± 848.8 | 0.4026 | 401.8 ± 1,161.7 | 244.5 ± 769.9 | 0.0746 |
| Median, IQR | 0 (0–1,555.0) | 0 (0–0) | | 0 (0–0) | 0 (0–0) | | 0 (0–0) | 0 (0–0) | |
| ED costs | | | | | | | | | |
| Mean, SD | 673.4 ± 890.0 | 239.1 ± 572.2 | <0.0001 | 577.8 ± 1,042.1 | 255.9 ± 570.0 | <0.0001 | 439.5 ± 711.5 | 253.4 ± 571.2 | 0.0034 |
| Median, IQR | 341.0 (0–958.0) | 0 (0–238.0) | | 0 (0–741.0) | 0 (0–274.0) | | 0 (0–675.0) | 0 (0–248.0) | |
| Cancer clinics costs | | | | | | | | | |
| Mean, SD | 19,332.4 ± 11,850.8 | 522.7 ± 7637.9 | 0.0902 | 9,963.8 ± 8640.4 | 442.2 ± 7318.0 | 0.1505 | 7,823.9 ± 9,453.7 | 482.4 ± 7,849.8 | 0.1614 |
| Median, IQR | 21,862.0 (10,789.0– 27,584.0) | 0 (0–0) | | 7,829.0 (5,832.0– 11,597.5) | 0 (0–0) | | 4,385.0 (1,674.0– 9,034.0) | 0 (0–0) | |
| Public drug plan (ODB) costs | | | | | | | | | |
| Mean, SD | 2,791.1 ± 3,981.5 | 1,723.9 ± 4,383.8 | 0.0014 | 2,747.5 ± 5,751.2 | 1,584.7 ± 3,251.9 | 0.0044 | 4,699.6 ± 16,895.9 | 1,811.3 ± 4,196.1 | 0.0043 |
| Median, IQR | 1,112.0 (555.0– 3,389.0) | 728.0 (2,14.5– 1,734.0) | | 934.5 (314.0– 2,355.5) | 695.0 (191.0– 1,683.0) | | 852.0 (285.0– 1,971.0) | 599.0 (166.0– 1,698.0) | |
| NDFP chemotherapy drug costs | | | | | | | | | |
| Mean, SD | 31,804.5 ± 20,979.9 | n.a. | 0.1575 | 12,871.7 ± 12,576.0 | n.a. | 0.1491 | 8,616.0 ± 11,849.0 | n.a. | 0.3159 |
| Median, IQR | 36,480.0 (13,042.0– 48,362.0) | n.a. | | 12,723.0 (0– 15,715.5) | n.a. | | 6,562.0 (0– 10,306.0) | n.a. | |
| Aggregated costs for other services ˆ | | | | | | | | | |
| Mean, SD | 8,731.1 ± 11,754.2 | 4,299.5 ± 13,807.9 | <0.0001 | 4,806.5 ± 5,550.9 | 4,071.9 ± 13,606.4 | 0.2573 | 5,344.7 ± 9,005.5 | 3,935.3 ± 11,220.9 | 0.1254 |
| Median, IQR | 6,187.0 (3,877.0– 9,006.0) | 787.5 (398.5– 1,815.5) | | 2,913.5 (1,940.0– 5,936.0) | 748.0 (422.0– 1,902.0) | | 2,746.0 (1,785.0– 5,527.0) | 650.0 (406.0– 1,605.0) | |

* Double suppression was conducted according to ICES reporting standards to reduce the risk of patient re-identification. ˆ Includes direct costs for dialysis clinics, rehabilitation services, complex and continuing care, home care services, OHIP lab billings, OHIP non-physician billings, OHIP shadow billings, FHO/FHN physician capitation, long-term care, OMHRS admissions, assisted devices, and outpatient hospital clinic visits. n.a.—not available.

### 3.4. Direct Healthcare Costs

The mean total healthcare cost per patient in the first year after diagnosis was estimated at CAD 77,554.7 ± 40,788.5 for the MCL cohort and CAD 11,123.8 ± 24,515.1 for the matched controls (Table 2). For MCL patients, the mean costs steadily decreased over the follow-up period, while the costs remained relatively the same from one year to the next in the control population. The costs associated with cancer clinics (mean: CAD 19,332.4 ± 11,850.8) and NDFP chemotherapy drugs (mean: CAD 31,804.5 ± 20,979.9) were the greatest contributors (24.9% and 41.0%, respectively) to the total healthcare costs for MCL patients in year one, which then decreased in years two and three. In the general population controls, costs for other services (38.7%) contributed most to the total healthcare costs in the first year

following index, followed by hospital costs (26.4%), a trend that continued throughout the follow-up period. Hospital costs include costs associated with inpatient hospitalizations, same-day surgeries, and ED visits. The average costs for MCL patients were higher than the controls for all healthcare categories, and these differences were statistically significant ($p < 0.05$), except for costs associated with NDFP and cancer clinics. Due to the small number of controls with any NDFP or cancer clinic costs, the models were less likely to detect differences between the cohort and controls.

### 3.5. Stratified Analysis

Overall, patients receiving BR as a first LoT had fewer healthcare encounters than patients receiving other therapies (Table 3). In the first year of follow-up, 18.0 visits to oncologists and hematologists, on average, were recorded for all MCL patients regardless of the first LoT. Th utilization of ED visits were also similar each year between the two groups. Throughout each year of follow-up, BR patients had fewer hospitalizations than patients on other treatments.

**Table 3.** Distribution of outcomes for matched MCL patients, according to LoT categories.

| BOI Outcomes | BR | | | Other | | |
|---|---|---|---|---|---|---|
| **Number of Patients** | **Year 1** | **Year 2** | **Year 3** | **Year 1** | **Year 2** | **Year 3** |
| | (N = 123) | (N = 112) | (N = * 101–105) | (N = 36) | (N = 28) | (N = * 22–26) |
| Person years | | | | | | |
| Mean, SD | 0.95 ± 0.18 | 0.95 ± 0.18 | 0.94 ± 0.18 | 0.90 ± 0.23 | 0.93 ± 0.18 | 0.92 ± 0.26 |
| Median, IQR | 1 (1–1) | 1 (1–1) | 1 (1–1) | 1 (1–1) | 1 (1–1) | 1 (1–1) |
| | | | | | | |
| **Healthcare resource utilization (HCRU)** | | | | | | |
| Attending physician specialty | | | | | | |
| Medical oncologist | 9 (7.3%) | * 6–10 | * 4–8 | 6 (16.7%) | * 1–5 | * 1–5 |
| Hematologist | 61 (49.6%) | 52 (46.4%) | * 31–35 | 12 (33.3%) | 6 (21.4%) | * 1–5 |
| Internal medicine | 18 (14.6%) | * 6–10 | * 5–9 | 6 (16.7%) | * 1–5 | * 1–5 |
| Family practice/GP | 23 (18.7%) | 34 (30.4%) | 42 (41.2%) | 10 (27.8%) | 12 (42.9%) | 14 (60.9%) |
| Other | * 9–13 | * 8–12 | * 7–11 | * 1–5 | * 1–5 | * 1–5 |
| No attending physician | 0 (0%) | * 1–5 | * 1–5 | 0 (0%) | 0 (0%) | * 1–5 |
| GP visits | | | | | | |
| Mean, SD | 12.0 ± 16.2 | 9.2 ± 9.8 | 10.8 ± 13.8 | 13.5 ± 14.0 | 12.4 ± 12.4 | 11.3 ± 9.5 |
| Median, IQR | 7 (5–14) | 6 (4–12) | 7 (3–12) | 11 (3–17) | 8 (5–17) | 10 (4–16) |
| Oncologist and hematologist visits | | | | | | |
| Mean, SD | 18.0 ± 14.1 | 10.7 ± 10.7 | 8.6 ± 10.0 | 18.0 ± 17.2 | 11.3 ± 12.9 | 6.7 ± 7.8 |
| Median, IQR | 19 (5–26) | 8 (5–13) | 5 (2–10) | 16 (6–22) | 7 (3–16) | 5 (2–8) |
| Other specialist visits | | | | | | |
| Mean, SD | 23.8 ± 18.1 | 15.6 ± 16.9 | 17.2 ± 18.9 | 38.2 ± 37.5 | 27.4 ± 31.5 | 22.3 ± 28.8 |
| Median, IQR | 19 (11–32) | 10 (5–20) | 12 (5–22) | 28 (13–45) | 14 (8–35) | 11 (5–27) |

**Table 3.** *Cont.*

| BOI Outcomes | BR | | | Other | | |
|---|---|---|---|---|---|---|
| Number of Patients | Year 1 | Year 2 | Year 3 | Year 1 | Year 2 | Year 3 |
| Inpatient hospitalizations | | | | | | |
|   Mean, SD | 0.7 ± 1.0 | 0.6 ± 1.1 | 0.5 ± 0.9 | 1.2 ± 1.5 | 0.9 ± 1.5 | 0.6 ± 0.9 |
|   Median, IQR | 0 (0–1) | 0 (0–1) | 0 (0–1) | 1 (0–2) | 0 (0–2) | 0 (0–1) |
| ED visits | | | | | | |
|   Mean, SD | 1.4 ± 1.8 | 1.1 ± 1.8 | 1.0 ± 1.5 | 1.6 ± 1.9 | 1.8 ± 2.5 | 1.1 ± 1.6 |
|   Median, IQR | 1 (0–2) | 0 (0–1) | 0 (0–1) | 1 (0–3) | 1 (0–3) | 0 (0–2) |
| **Direct healthcare costs** | | | | | | |
| Total direct costs | | | | | | |
|   Mean, SD | 79,302.5 ± 41,021.2 | 39,424.3 ± 25,727.5 | 37,340.9 ± 37,134.5 | 71,583.1 ± 39,968.5 | 42,767.9 ± 38,934.2 | 30,375.2 ± 32,497.0 |
|   Median, IQR | 87,738.0 (51,445.0–109,025.0) | 31,123.0 (25,055.0–52,479.0) | 24,557.0 (15,091.0–45,068.0) | 59,583.0 (50,019.0–81,740.0) | 32,051.0 (24,349.0–56,171.0) | 21,001.0 (13,126.0–33,795.0) |
| GP costs | | | | | | |
|   Mean, SD | 450.9 ± 816.0 | 380.9 ± 686.4 | 485.8 ± 996.8 | 504.7 ± 787.9 | 665.0 ± 1,165.6 | 428.6 ± 535.4 |
|   Median, IQR | 191.0 (19.0–515.0) | 174.0 (11.0–430.0) | 122.0 (10.0–435.0) | 173.0 (25.0–701.0) | 182.0 (21.0–661.0) | 215.0 (14.0–694.0) |
| Oncologist and hematologist costs | | | | | | |
|   Mean, SD | 932.8 ± 837.5 | 324.9 ± 418.5 | 290.6 ± 443.9.0 | 818.0 ± 769.8 | 259.4 ± 298.8 | 169.1 ± 152.0 |
|   Median, IQR | 802.0 (91.0–1,580.0) | 203.0 (0–505.0) | 71.0 (0–327.0) | 899.0 (17.0–1,337.0) | 149.0 (0–395.0) | 165.0 (34.0–292.0) |
| Other specialist costs | | | | | | |
|   Mean, SD | 3,113.2 ± 2,256.4 | 1,995.0 ± 2,149.9 | 2,169.6 ± 2,521.1 | 4,156.6 ± 3,366.8 | 2,985.3 ± 3,321.1 | 3,090.1 ± 3,185.6 |
|   Median, IQR | 2,605.0 (1,664.0–3,932.0) | 1,231.0 (532.0–2,735.0) | 1,247.0 (553.0–2,458.0) | 3,386.0 (2,341.0–4,634.0) | 1,064.0 (716.0–4,802.0) | 1,844.0 (718.0–4,029.0) |
| Inpatient hospitalization costs | | | | | | |
|   Mean, SD | 6,483.9 ± 10,150.0 | 4,781.0 ± 12,581.3 | 5,617.7 ± 13,803.6 | 17,391.7 ± 34,333.4 | 11,064.0 ± 25,587.3 | 7,458.7 ± 14,121.9 |
|   Median, IQR | 0 (0–10,135.0) | 0 (0–4,678.0) | 0 (0–4,189.0) | 4,646.0 (0–19,108.0) | 0 (0–15,813.0) | 0 (0–12,644.0) |

**Table 3.** *Cont.*

| BOI Outcomes | BR | | | Other | | |
|---|---|---|---|---|---|---|
| Number of Patients | Year 1 | Year 2 | Year 3 | Year 1 | Year 2 | Year 3 |
| Same-day surgery costs | | | | | | |
| Mean, SD | 892.9 ± 1,114.1 | 471.1 ± 1,347.2 | 438.6 ± 1,254.7 | 658.6 ± 1,287.8 | n.a. | n.a. |
| Median, IQR | 0 (0–1,690.0) | 0 (0–460.0) | 0 (0–0) | 0 (0–837.0) | n.a. | n.a. |
| ED costs | | | | | | |
| Mean, SD | 632.0 ± 857.9 | 469.5 ± 859.5 | 429.5 ± 707.1 | 814.9 ± 991.7 | 1,011.0 ± 1,519.9 | 484.0 ± 745.4 |
| Median, IQR | 340.0 (0–938.0) | 0 (0–621.0) | 0 (0–687.0) | 512.0 (0–1,481.0) | 395.0 (0–1,670.0) | 0 (0–675.0) |
| Cancer clinics costs | | | | | | |
| Mean, SD | 20,942.5 ± 12,340.3 | 9,841.0 ± 7,866.4 | 8,052.1 ± 9,535.7 | 13,831.3 ± 7,933.6 | 10,454.8 ± 11,385.8 | 6,812.1 ± 9,218.1 |
| Median, IQR | 25,106.0 (11,707.0–29,167.0) | 8,249.0 (5,860.0–11,535.0) | 5,474.0 (1,957.0–9,921.0) | 14,022.0 (8,595.0–17,864.0) | 7,688.0 (2,210.0–15,774.0) | 3,914.0 (0–8,647.0) |
| ODB costs | | | | | | |
| Mean, SD | 2,210.5 ± 3,147.2 | 2,708.7 ± 6,205.2 | 5,246.0 ± 18,568.3 | 4,774.9 ± 5,639.6 | 2,902.5 ± 3,457.8 | 2,276.7 ± 4,316.9 |
| Median, IQR | 1,000.0 (465.0–2,488.0) | 872.0 (267.0–2,191.0) | 859.0 (285.0–2,026.0) | 1,755.0 (896.0–8,319.0) | 1,375.0 (438.0–4,062.0) | 737.0 (266.0–1,887.0) |
| NDFP chemotherapy drug costs | | | | | | |
| Mean, SD | 35,590.8 ± 21,638.1 | 14,180.9 ± 13,138.7 | 9,716.0 ± 12,695.9 | 18,868.2 ± 11,508.0 | 7,635.0 ± 8,305.6 | 3,737.7 ± 4,566.0 |
| Median, IQR | 43,185.0 (17,189.0–52,399.0) | 13,136.0 (6,075.0–15,850.0) | 7,032.0 (0–11,752.0) | 21,396.0 (8,384.0–26,922.0) | 4,864.0 (0–14,540.0) | 0 (0–8,320.0) |
| Aggregated costs for other services ˆ | | | | | | |
| Mean, SD | 8,349.0 ± 12,665.6 | 4,519.8 ± 5,536.0 | 5,158.4 ± 8,755.6 | 10,036.7 ± 7,890.4 | 5,953.4 ± 5,561.2 | 6,170.7 ± 10,212.1 |
| Median, IQR | 5,638.0 (3,638.0–8,019.0) | 2,748.0 (1,853.0–5,241.0) | 2,781.0 (1,748.0–5,409.0) | 7,996.0 (4,886.0–11,106.0) | 3,550.0 (2,305.0–8,771.0) | 2,735.0 (1,839.0–6,148.0) |

* Double suppression was conducted according to ICES reporting standards to reduce the risk of patient re-identification. ˆ Includes direct costs for dialysis clinics, rehabilitation services, complex and continuing care, home care services, OHIP lab billings, OHIP non-physician billings, OHIP shadow billings, FHO/FHN physician capitation, long-term care, OMHRS admissions, assisted devices, and outpatient hospital clinic visits. n.a.—not available.

In the first year following diagnosis, MCL patients on BR incurred higher total costs (mean: CAD 79,302.5 ± 41,021.2) than patients on other regimens (mean: CAD 71,583.1 ± 39,968.5). The costs for both groups in years two and three were approximately half of those in year one, while the costs associated with patients on other regimens were higher than the costs incurred by patients on BR in the second year of follow-up (Table 3). For those who received BR, NDFP chemotherapy drug costs were the greatest contributor to the total healthcare expenditures across all three years of follow-up (26.0–44.9%). On the

other hand, hospital costs contributed the most (26.1–28.2%) to the total healthcare costs for patients on other regimens as a first LoT.

### 3.6. Survival Outcomes and Time to Next Treatment or Death

For patients in the MCL cohort, mortality in the three-year period was 31.4% (95% CI: 24.4–38.8%) (Figure 1A). There was a statistically significant difference ($p < 0.05$) in survival between patients who received BR vs. other regimens as a first LoT, with mortality in the third year estimated at 27.6% (95% CI: 20.0–35.8%) and 44.4% (95% CI: 27.7–59.9%), respectively (Figure 1B).

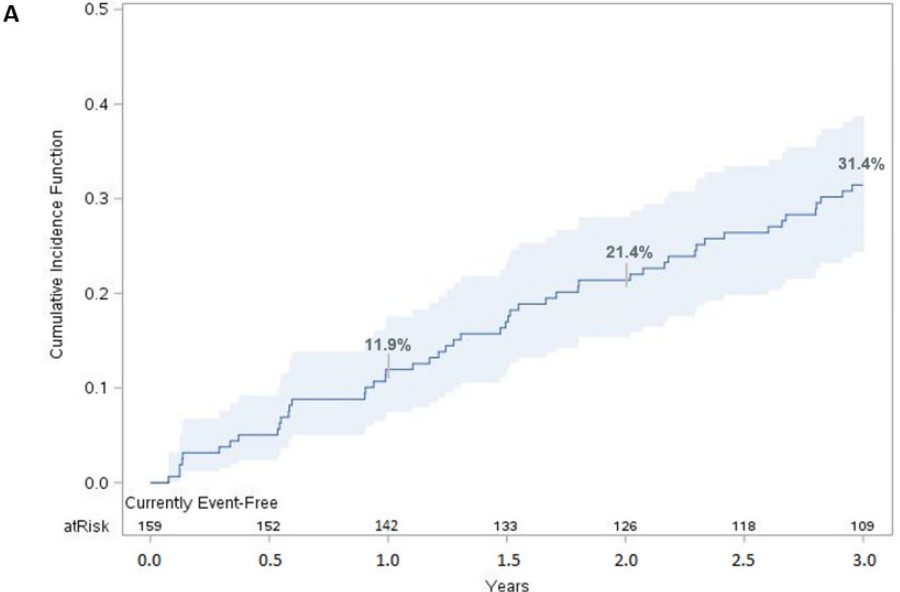

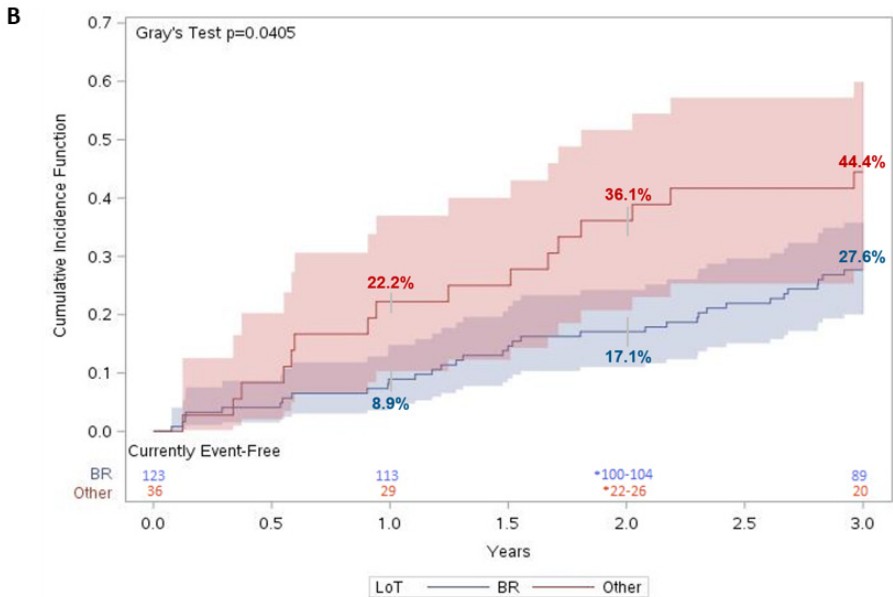

**Figure 1.** Three-year overall survival for MCL patients. Cumulative incidence function of all-cause mortality for (**A**) matched MCL patients (n = 159) and (**B**) matched MCL patients stratified according to LoT. For each time t, patients were considered "at risk" if they had not had an event of interest before time t and were not censored before or at time t. Number at risk was evaluated on day 365 for year 1, day 730 for year 2, and day 1095 for year 3. Shaded regions represent 95% confidence interval (CI).

Three years after the initiation of a first LoT, 40.9% (95% CI: 33.2–48.4%) of the MCL patients initiated a second-line treatment or died (Figure 2A). For patients who initiated a second LoT, the median time to the second LoT was 15.2 months (1.3 years). When stratified by regimen received, only 35.8% (95% CI: 27.4–44.2%) of patients on BR had a combined outcome of initiating a second LoT or death three years after initiating a first LoT, compared with 58.3% (95% CI: 40.2–72.7%) of patients who received another regimen (Figure 2B). The difference in TTNTD was statistically significant ($p < 0.005$).

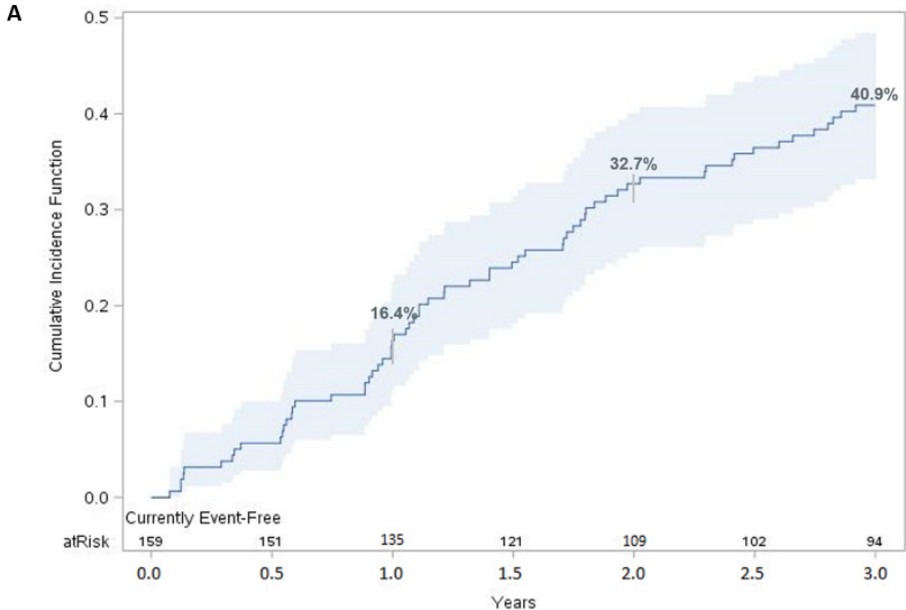

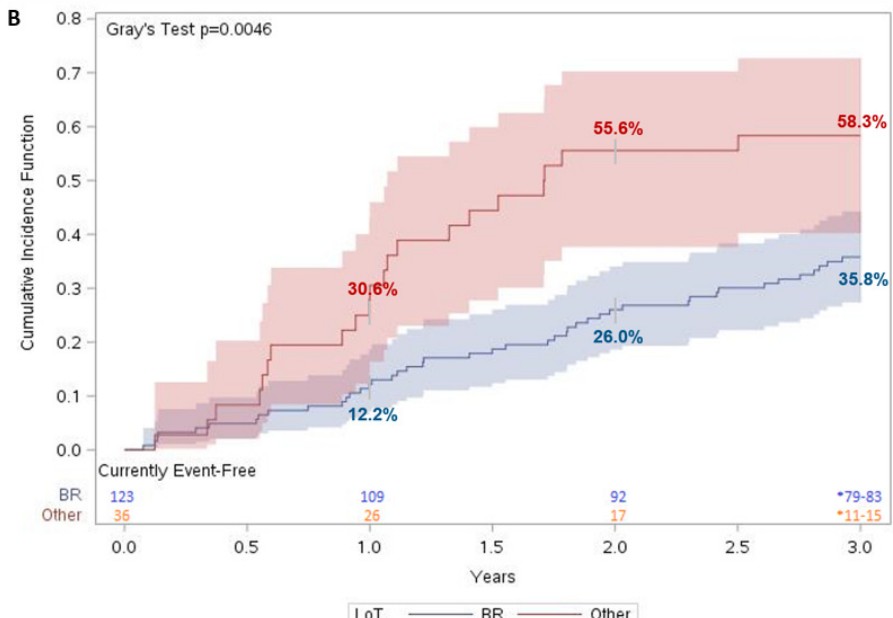

**Figure 2.** Time to next therapy or death for MCL patients. Cumulative incidence function of TTNTD for (**A**) matched MCL patients (n = 159) and (**B**) matched MCL patients stratified by LoT. For each time t, patients were considered "at risk" if they had not had an event of interest before time t and were not censored before or at time t. Number at risk was evaluated on day 365 for year 1, day 730 for year 2, and day 1095 for year 3. Shaded regions represent 95% confidence interval (CI).

## 4. Discussion

This study is one of the first to use real-world Canadian data to study MCL patients and describe their BOI. The availability of retrospective administrative data in Ontario has allowed us to follow newly diagnosed MCL patients and track their HCRU, costs, and outcomes, all of which were stratified by first LoT received. Overall, MCL patients aged $\geq$65 had higher HCRU and costs compared to their matched general population controls in all three years following diagnosis. Both healthcare utilization and costs were highest in the first year following diagnosis for the MCL cohort, which then decreased in subsequent years. A significantly lower number of MCL patients who received BR as a first LoT died or started a second line within three years compared to patients who received another regimen. This difference may be because more intense regimens (e.g., R-CHOP) may have been given to patients with a more aggressive disease; hence, the analysis of TTNTD using a composite outcome may mask the difference in the time to receive the next line of therapy and that to the time of death. This finding highlights the inherent variability of the patient cohort (Table 1). The variability may also contribute to the wide distribution of costs observed in this study, as similarly observed in previous retrospective cohort studies using administrative data [19,20].

Previous estimates of the burden of illness for MCL have highlighted that adverse events are key drivers of increased costs and resource use [14]. Hepatoxicity, stroke, and renal failure were the AEs associated with the highest medical costs [21]; however, these findings do not align with the Canadian experience. High-dose chemotherapy was found to be more resource intensive relative to chemoimmunotherapy, consistent with the present findings of NDFP chemotherapy costs being major contributors to the significantly higher healthcare costs for the MCL cohort. Consistently, another study found that the increased medical costs incurred by MCL patients in China were highly sensitive to the drug acquisition costs of rituximab [22]. While BR was also found by another US study [21] to be the most common therapy for MCL patients, inpatient admissions and office visits were found to be the main cost drivers for patients treated with BR [14]. However, this inconsistency in cost drivers may be attributable to the variabilities in health system processes between the USA and Canada [14].

The NDFP pays for newer and high-cost injectable cancer drugs in Ontario that are administered in hospitals and cancer centers [23]. While the mean NDFP chemotherapy drug costs for the MCL cohort ranged between CAD 9000–32,000 during the first three years following diagnosis, these NDFP costs are relatively low compared to the NDFP costs incurred by patients of other cancer types [24,25]. In Canada, biosimilars for rituximab were approved for the treatment of NHL in 2019 [26]. As the majority of patients in this study were followed until a period prior to 2019, the current costs associated with MCL treatment are likely to have been reduced with the use of biosimilar rituximab in clinical practice. However, as biosimilars for rituximab are only administered intravenously, there is a cost trade-off between the increased patient chair time and the active healthcare professional time compared to when the subcutaneous formulation of innovator rituximab is used [27].

There is limited evidence of the economic burden of MCL in the literature, most of which has focused on costs according to the treatment regimen and care setting. Because Ontario has a single-payer system for medical claims, the data available for this study are census-based and thus generalizable and applicable for policy, presenting a major strength of the study. This study also had some limitations. First, the sample for this analysis constituted 50.8% of all MCL patients newly diagnosed between 1 January 2013 and 31 December 2016. The majority (N = 100) of patients were excluded as they did not have a record of systemic therapy in the ALR within three years from index date or before 31 December 2019. However, the number of patients excluded aligns with previous estimates, which highlighted that up to 30% of MCL patients have a comparably indolent course even while untreated ("watchful waiting") [28]. Restricting the analyses to only those who received systemic therapy may impact the generalizability of our results to a broader population of individuals who are diagnosed with MCL but do not commence

systemic therapy within three years following diagnosis. Second, the administrative data used in this study only include publicly reimbursed medical and prescription drug claims; therefore, cash, private claims, and patient support programs are not captured in this study. Furthermore, administrative data are a type of secondary data source wherein a lag in the availability of the most recent data subsists. For example, at the time of study initiation, the OCR contained cancer diagnoses up until 31 December 2020, with data refreshes dependent on the data-sharing agreements between ICES and Ontario Health. As such, the most recent trends for MCL patients may not be captured.

The treatment landscape for MCL is evolving, with encouraging responses to targeted therapy (e.g., Bruton's tyrosine kinase inhibitors (BTKi), venetoclax) in relapsed/refractory MCL patients, leading to its use in the relapsed setting (non-curative), as well as in exploratory clinical trials for treatment-naïve MCL patients [5,29,30]. Other emerging therapies such as CAR-T cell therapy and bispecific antibody therapy are also showing promising efficacy and may improve survival outcomes for patients with progressive MCL following treatment with a BTKi [29,31].

MCL patients often relapse after first-line chemoimmunotherapy [32], with many patients requiring subsequent lines of treatment. However, MCL mostly occurs in older individuals, for whom intense chemotherapy is generally not advised due to higher risks of chemotherapy-related toxicities [33]. The likelihood of accumulating related medical problems that lead to hospitalization also increases with the repeated use of chemotherapy. These challenges highlight the need for targeted therapies, such as BTKis, for the treatment of MCL; their incorporation into the first LoT is currently being explored [29]. In the meantime, BR remains the preferred option among patients aged $\geq$65 as it is considered less toxic. Our study demonstrated that those who initiated BR as a first LoT incurred lower HCRU and costs, and also experienced longer TTNTD and 3-year OS compared to those who received another regimen.

## 5. Conclusions

This study provides a comprehensive estimate of healthcare resource utilization and the costs associated with the treatment of newly diagnosed MCL in individuals aged $\geq$65. This study found that the management of newly diagnosed MCL patients represents a significant burden on the healthcare system, driven by chemoimmunotherapy and cancer clinic costs. BR was found to be the most common therapy in first LoT, which is the standard of care in many settings [6]. Although associated with greater total costs, patients who received BR were observed to experience better overall survival and TTNTD than those on other regimens. Future research should look at the same outcomes with emergent therapy as more administrative data accrue.

**Supplementary Materials:** The following supporting information can be downloaded at: https://www.mdpi.com/article/10.3390/curroncol30060418/s1, Table S1: Regimens considered as first-line therapy for MCL; Table S2: Patient selection into MCL cohort, Ontario, Canada, 2013–2016; Table S3: First line therapy received by the MCL cohortˆ; Table S4: Demographic and clinical characteristics for matched MCL patients, by LoT categories.

**Author Contributions:** Conceptualization, P.A., J.E.-P., S.M., A.K., A.S. and E.M.E.; methodology, P.A., J.E.-P., M.E., S.M., A.K., A.S. and E.M.E.; validation, M.E. and S.M.; formal analysis, M.E. and S.M.; investigation, M.E. and S.M.; resources, A.K. and A.S.; writing—review and editing, P.A., J.E.-P., M.E., S.M., A.K., A.S. and E.M.E.; supervision, J.E.-P., A.K., A.S. and E.M.E.; project administration, A.K. and A.S; funding acquisition, J.E.-P. and E.M.E. All authors have read and agreed to the published version of the manuscript.

**Funding:** This research was funded by Janssen Inc.

**Institutional Review Board Statement:** The study was conducted in accordance with the Declaration of Helsinki and approved by the ICES Privacy and Compliance Office. Advarra IRB# 00000971, Approval #IBR-C-21-CAN-002-V01/2722439.

**Informed Consent Statement:** The use of data in this project was authorized under Section 45 of Ontario's Personal Health Information Protection Act and, as a result, informed consent was not required.

**Data Availability Statement:** The data from this study is held securely in coded form at ICES. While legal data sharing agreements between ICES and data providers (e.g., healthcare organizations and the government) prohibit ICES from making the dataset publicly available, access may be granted to those who meet pre-specified criteria for confidential access, available at www.ices.on.ca/DAS (accessed on 5 June 2023) (email das@ices.on.ca). The full dataset creation plan and underlying analytic code are available from the authors upon request, understanding that the computer programs may rely upon coding templates or macros that are unique to ICES and are, therefore, either inaccessible or may require modification.

**Acknowledgments:** Thank you to Keresa Arnold, Jacob Etches, and Bo Zhang of ICES for data management and analytic support. This study made use of de-identified data from the ICES Data Repository, which is managed by ICES with support from its funders and partners: Canada's Strategy for Patient-Oriented Research (SPOR), the Ontario SPOR Support Unit, the Canadian Institutes of Health Research, and the Government of Ontario. The opinions, results, and conclusions reported are those of the authors. No endorsement by ICES or any of its funders or partners is intended or should be inferred. We acknowledge support from Asad Husain and Angeline Zhu of Janssen Inc. for strategic input into this study. Medical writing support was provided by Ceryl Tan of IQVIA.

**Conflicts of Interest:** Peter Anglin is a stockholder of Johnson & Johnson. Julia Elia-Pacitti is an employee of Janssen Inc. and a stockholder of Johnson & Johnson. Maria Eberg, Sergey Muratov, Atif Kukaswadia, and Arushi Sharma are employees of IQVIA Solutions Canada Inc. IQVIA is a contract research organization that received consulting fees from Janssen Inc. Emmanuel M. Ewara is an employee of Janssen Inc. and a stockholder of Johnson & Johnson.

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
