# Peer review of "Estimating the Associated Burden of Illness and Healthcare Utilization of Newly Diagnosed Patients Aged ≥65 with Mantle Cell Lymphoma (MCL) in Ontario, Canada"

_curroncol, doi:10.3390/curroncol30060418_

Round 1

Reviewer 1 Report

This is an excellent study examining the real-world healthcare utilization and costs associated with MCL in a matched cohort study of 159 older patients over a 3 year period post-diagnosis. Major findings include the breakdown of physician visits and other service usage and the estimation of costs, averaging a total of 78K in the first year, and then approx. halving to 36-40K each year in the latter 2 years. Survival rates are reported overall and also by treatment and bendamustine+rituximab (BR) as a first line of treatment was associated with better survival. Surprisingly GP visits were fairly consistent over the 3 year period which suggests the importance of communication and collaboration with this frontline. The authors were careful in their interpretations, noting that although BR was associated with survival, this may be hard to conclude a causal connection given the study design- nonetheless the clinical protocol of using BR as first line when possible is supported. The authors were also thorough in pointing out the nature of the loss of 100 patients from the initial cohort due to lack of medical records and to explain that the lack of significant difference in cancer clinic costs may be attributable to the very small sample of cancer patients in the control group. Overall I have no further suggestions for improvement, aside from the minor issue with the formatting of the citation numbers in the main text.

Author Response

We thank the reviewer for their feedback and support for the manuscript. The formatting issue with the citation numbers in the main text has been addressed.

Reviewer 2 Report

Authors have provided the comprehensive estimates of health care resource utilization and the expense associated with the treatment new diagnosed MCL in elder patients in this manuscript. Overall the study has been planned, analyzed and compiled well. I do not have any specific comments apart from few grammatical and English correction.

Author Response

We thank the reviewer for their feedback and support for the manuscript. We have corrected grammatical mistakes throughout the manuscript.

Reviewer 3 Report

The current article is quite relevant in perspective to the cost so incurred while diagnosing cancer. Author has made tried to touched the domain which may clearly justify the title so mentioned. But, I think, without the relevant biological experimental set up, the current oncology and is domain is still remained incomplete. However, the title of the manuscript is well justified with section so listed. Here, I have listed few relevant comments that may be addressed to make its more conclusive.

1.    1. Author must mention the technique used for the diagnosis for MCL in study design. What marker or gene used for MCL.

2.    2.  If possible, it is suggested to add the data for patient with age ranges between 40-65 years. It may provide the age specific differences in the cost for patients.

3.     3. Author may add a section suggesting the major or an approach for overcoming or lowering such cost so incurred. It will ensure the manuscript more conclusive.

Author Response

We thank the reviewer for their comments. Please see below for our response to each comment raised:

  1. As this was a retrospective, population-based study, the diagnosis of MCL has previously been made by the respective physician responsible for each patient included in the cohort. We therefore identified patients with an MCL diagnosis using the International Classification of Disease for Oncology (ICD-O-3) morphology code for MCL (96733), which is detailed in the Methods section (line 77). We do not have information on which marker or gene was used to diagnose MCL for the patients included in the cohort.
  2. The focus of our study is on newly diagnosed Canadian patients aged ≥65 as the median age at which patients are diagnosed with MCL ranges from 60 – 70 years old (PMID: 30725670). In addition, as patients ≥65 years of age are eligible for Ontario’s publicly funded drug program, we felt that focusing our analysis to this age group allowed for a more fulsome perspective of healthcare costs and utilization. We also selected patients ≥65 years of age at index as this is the usual cutoff for transplant eligibility. While we agree that comparing the differences in the cost incurred by patients aged 40-65 and ≥65 may be interesting, it is outside the scope of this present study.
  3. The study found NDFP chemotherapy drug costs to be the greatest contributor to the total healthcare costs associated with MCL treatment. Since biosimilars for rituximab were introduced in Canada 3 years after the end of our study period, current cost estimates may be lower than what was quantified in the present study. This has been detailed in the Discussion section (line 310).

Round 2

Reviewer 3 Report

Dear Author,

Thanks for making changes in the Article. Though, still rectification was demanded.

Moreover, It is fine.

Thanks.

Manish

Author Response

We thank the reviewer for their comments. We have added a paragraph on potential approaches to reduce costs associated with MCL treatment.